# Physically Compatible 3D Object Modeling from a Single Image

**Minghao Guo**[1], **Bohan Wang**[1*], **Pingchuan Ma**[1], **Tianyuan Zhang**[1], **Crystal Elaine Owens**[1],
**Chuang Gan**[2,3], **Joshua B. Tenenbaum**[1,4,5], **Kaiming He**[1], **Wojciech Matusik**[1]

[1]MIT CSAIL, [2]UMass Amherst, [3]MIT-IBM Waston AI Lab, [4]MIT BCS,
[5]Center for Brains, Minds and Machines

https://gmh14.github.io/phys-comp/

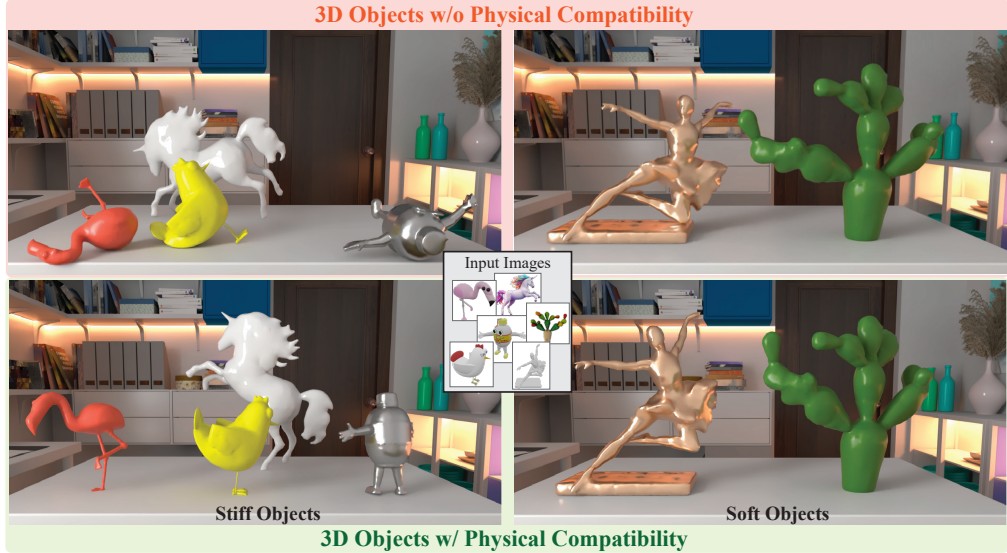

Figure 1: Existing methods for single-view reconstruction often result in objects that, when subjected to real-world physical forces (such as gravity) and user-required mechanical materials, exhibit problematic behaviors such as toppling over (top left) and undesirable deformation (top right), diverging from their intended depiction in the input images. In contrast, our approach produces physical objects that maintain stability (bottom left) and mirror the objects' static equilibrium state captured in the input images (bottom right).

## Abstract

We present a computational framework that transforms single images into 3D physical objects. The visual geometry of a physical object in an image is determined by three orthogonal attributes: mechanical properties, external forces, and rest-shape geometry. Existing single-view 3D reconstruction methods often overlook this underlying composition, presuming rigidity or neglecting external forces. Consequently, the reconstructed objects fail to withstand real-world physical forces, resulting in instability or undesirable deformation – diverging from their intended designs as depicted in the image. Our optimization framework addresses this by embedding physical compatibility into the reconstruction process. We explicitly decompose the three physical attributes and link them through static equilibrium, which serves as a hard constraint, ensuring that the optimized physical shapes exhibit desired physical behaviors. Evaluations on a dataset collected from Objaverse demonstrate that our framework consistently enhances the physical realism of 3D models over existing methods. The utility of our framework extends to practical applications in dynamic simulations and 3D printing, where adherence to physical compatibility is paramount.

---

[*]Corresponding author.

38th Conference on Neural Information Processing Systems (NeurIPS 2024).

# 1 Introduction

The field of single-image 3D shape modeling has experienced significant advancements over the past years, largely propelled by advances in single-view reconstruction techniques. These methods, ranging from generating multi-view consistent images for per-scene 3D reconstruction [22, 23, 21, 24, 32, 20], to employing large reconstruction models (LRMs) for feedforward inference [12, 41, 44, 50, 47, 40], have enhanced the geometric quality and visual fidelity of the 3D shapes to unprecedented levels.

However, reconstructing a 3D shape from an image often aims to be beyond a mere visualization. These generated objects find applications in virtual environments such as filming and gaming, as well as in tangible fields like industrial design and engineering. Despite their diverse applications, a common oversight in many current single-view reconstruction methods is the negligence of physical principles. As shown in the top row of Fig. 1, when subjected to real-world physics such as gravity, these 3D objects produced from these techniques exhibit issues such as instability and undesired deformation, diverging from their depiction in the input images. Such inconsistency can significantly undermine the practical utility of the models, as they fail to meet the functional and aesthetic expectations set by the original image.

Fundamentally, an image is more than a visual representation of an object: It captures a physical snapshot of the object in a state of static equilibrium, under the influence of real-world forces. In this context, the geometry seen in an image is determined by three orthogonal attributes: *mechanical properties*, *external forces*, and *rest-shape geometry*. As shown in the inset figure, these attributes collectively model the spectrum of potential static configurations that a physical object might adopt. Reconstructing such an object from an image is essentially an ill-posed problem, since multiple combinations of these attributes can result in identical static geometry. Current methods, however, often overlook this underlying composition; they typically assume objects are rigid or neglect the impact of external forces. The reconstructed objects thus merely replicate the visual geometry without considering the three physical attributes.

To bridge this gap, we explicitly decompose these attributes for reconstructing a physical object from a single image. Our framework holistically takes mechanical properties and external forces as predefined inputs, reflecting typical user specifications in real-world applications like 3D printing and simulations. The output is the rest-shape geometry tailored to these inputs. These attributes are integrated through the principles of static equilibrium physics. This explicit decomposition imposes two stringent physical constraints in object modeling: static equilibrium is enforced as a *hard constraint*, and the physical object must conform to user-specified material properties and external forces. These resulting physical objects are stable, robust under real-world physics, and are high-fidelity replicas inferred from the input images, as shown in the bottom row of Fig. 1.

More specifically, we propose *physical compatibility* optimization, which is a physically constrained optimization with rest-shape geometry as the variable. In this setup, the objective is for the modeled physical object to exhibit desired behaviors, such as matching the geometry depicted in the input image under external forces and maintaining stability under gravity. The constraint is the equation of static equilibrium simulation, ensuring that during optimization, the physical object remains in the equilibrium state, with internal forces generated by deformation from the rest shape balancing the external forces. We parameterize the rest-shape geometry using a plastic deformation field and solve this hard-constrained optimization problem by using implicit differentiation with gradient descent.

For evaluation, we introduce five metrics designed to comprehensively assess the physical compatibility of the modeled 3D objects under simulation. These metrics include image loss between the rendered image of the modeled physical object and the input image, stability under gravity, as well as measures from finite element analysis, such as integrity and structural robustness. Our framework's versatility is demonstrated by its integration with five distinct single-view reconstruction methods, each employing unique geometry representations. Results on a dataset collected from Objaverse [9], consisting of 100 shapes, show that our framework consistently produces 3D objects with enhanced physical compatibility. Furthermore, we demonstrate the practical utility of our framework through applications in dynamic simulations and 3D printing fabrication.

## 2 Related Work

**Single-view 3D reconstruction.** Recent strides in single-view 3D reconstruction have mainly been fueled by data-driven methods, paralleled by advancements in 3D geometry representation, including NeRF [27], NeuS [43], triplanes [33], Gaussian splatting [16], surface meshes [29], and tet-spheres [11]. These developments have significantly enhanced the geometric quality and visual fidelity of the reconstructed 3D shapes. There are primarily two types of single-view reconstruction methods: 1) Test-time optimization-based methods [31, 23, 39, 45], use multiview diffusion models [21] and iteratively reconstruct 3D scenes using these diffusion priors. 2) Feedforward methods [13, 48, 38, 7, 44, 50] leverage large datasets and learn general 3D priors for shape reconstruction to enable efficient one-step 3D reconstruction from single or sparse views. Unlike the aforementioned methods, our work emphasizes the integration of physical modeling into the reconstruction process. This integration distinguishes our work by ensuring that the resulting 3D shapes are not only visually accurate but also physically plausible under real-world conditions.

**Physics-based 3D modeling.** There has been an increasing interest in incorporating physics into 3D shape modeling. While many approaches utilize video input, which offers a richer temporal context for inferring physical properties such as material parameters [51] and geometry [19], others approach the problem by first reconstructing an object's geometry from multi-view images and subsequently applying physical simulations [10, 46, 26, 25]. Additionally, several studies have explored extracting physical information from static images [49, 3, 37], using data-driven techniques to estimate properties like shading, mass, and material. In contrast, our work incorporates physical principles, specifically static equilibrium, as hard constraints within the reconstruction process. This integration allows for the optimization of 3D models that adhere to desired physical behaviors depicted by the image.

**Fabrication-aware shape design.** Originating from the computer graphics community, fabrication-aware shape design systems enable designers to specify higher-level objectives – such as structural integrity, deformation, and appearance – with the final shape as the output of the computational system [4]. Related methodologies in this domain, particularly those addressing static equilibrium, include inverse elastic shape design [8] and sag-free initialization [14]. However, these approaches typically require a manually created initial geometry, whereas our work aims to construct the physical object directly from a single input image.

## 3 Approach

Our objective is to create 3D objects from a single image that are physically compatible, ensuring that they align with the input image in the static equilibrium state while also meeting the stability requirements. Governed by universal physical principles, the physical behavior of an object is determined by its mechanical properties, external forces, and rest-shape geometry. Our framework treats the rest-shape geometry as the optimization variable, assuming that the mechanical properties and external forces are predefined as inputs. Fig. 2 illustrates the overall pipeline.

### 3.1 Formulation of Physical Compatibility

In our approach, we treat the entity depicted in the input image as a solid object. We employ Finite Element Method (FEM) for robust solid simulation. The object is represented by a volumetric mesh, denoted as $\mathcal{M} = (\mathbf{x}, \mathbf{T})$. Here, $\mathbf{x} \in \mathbb{R}^{3N}$ represents the 3D positions of the vertices, with $N$ denoting the total number of vertices. $\mathbf{T} \in \mathbb{N}^{Z \times K}$ describes the mesh connectivity, where $Z$ represents the total number of elements and $K$ indicates the number of vertices per element. The mesh in its *rest-shape geometry*, which is the state without any internal or external forces applied, is represented as $\mathcal{M}_{\text{rest}} = (\mathbf{X}_{\text{rest}}, \mathbf{T})$. The input image depicts the *static geometry*, which is the deformed geometry of the object under static equilibrium[1], denoted as $\mathcal{M}_{\text{static}} = (\mathbf{x}_{\text{static}}, \mathbf{T})$. In accordance with Newton's laws, $\mathbf{x}_{\text{static}}$ adheres to the following equation:

$$\mathbf{f}_{\text{int}}(\mathbf{x}_{\text{static}}, \mathbf{X}_{\text{rest}}; \Theta) = \mathbf{f}_{\text{ext}}(\mathbf{x}_{\text{static}}), \tag{1}$$

---

[1] Although our implementation employs *quasi-static equilibrium*, we use the term *static equilibrium* across the paper for consistency.

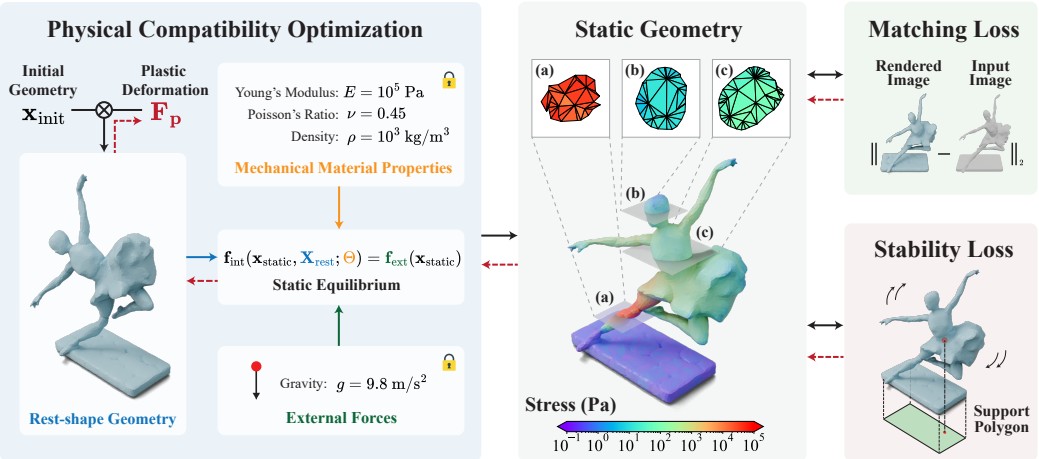

Figure 2: Overall pipeline. Given predefined mechanical properties and external forces, our pipeline optimizes the rest-shape geometry to ensure that the shape, when in a state of static equilibrium, aligns with the target image and meets stability criteria. We visualize the stress distribution of the static geometry using a colored heat map, illustrating the spatially varying deformation of the physical object under static equilibrium.

where $\mathbf{f}_{\text{int}}(\cdot, \cdot; \Theta) : \mathbb{R}^{3N} \times \mathbb{R}^{3N} \to \mathbb{R}^{3N}$ denotes the internal forces exerted by deformed objects transitioning from $\mathbf{X}_{\text{rest}}$ to $\mathbf{x}_{\text{static}}$, $\mathbf{f}_{\text{ext}}(\cdot) : \mathbb{R}^{3N} \to \mathbb{R}^{3N}$ embodies the external interaction forces such as gravity, and $\Theta$ represents the mechanical material properties, such as the stiffness of the object. Eq. 1 reveals that $\Theta$ (mechanical properties), $\mathbf{f}_{\text{ext}}$ (external forces), and $\mathbf{X}_{\text{rest}}$ (the rest-shape geometry) collectively determine the static geometry $\mathbf{x}_{\text{static}}$.

Given $\Theta$ and $\mathbf{f}_{\text{ext}}(\cdot)$, the goal of physically compatible modeling is to ensure that the rest-shape geometry $\mathcal{M}_{\text{rest}}$ conforms to given objectives under static equilibrium. This is formulated as the following optimization problem:

$$\min_{\mathbf{X}_{\text{rest}}, \mathbf{x}_{\text{static}}} \mathcal{J}(\mathbf{X}_{\text{rest}}, \mathbf{x}_{\text{static}}) = \mathcal{L}(\mathbf{x}_{\text{static}}) + \mathcal{L}_{\text{reg}}(\mathbf{X}_{\text{rest}})$$
$$\text{s.t.} \quad \mathbf{f}_{\text{int}}(\mathbf{x}_{\text{static}}, \mathbf{X}_{\text{rest}}; \Theta) = \mathbf{f}_{\text{ext}}(\mathbf{x}_{\text{static}}). \tag{2}$$

Here, $\mathcal{J}(\mathbf{X}_{\text{rest}}, \mathbf{x}_{\text{static}})$ is the objective function, consisting of $\mathcal{L}(\mathbf{x}_{\text{static}})$, which measures the alignment of the geometry $\mathbf{x}_{\text{static}}$ with the specified target. $\mathcal{L}_{\text{reg}}(\mathbf{X}_{\text{rest}})$ regularizes the rest-shape geometry $\mathbf{X}_{\text{rest}}$, with more details discussed in Section 3.2.

Within the scope of this work, two tasks for $\mathcal{L}(\mathbf{x}_{\text{static}})$ are considered: 1) $\mathbf{x}_{\text{static}}$ replicates the geometry depicted in the input image; and 2) $\mathbf{x}_{\text{static}}$ maintains stability and inherently remains upright without toppling. In the first scenario, the loss function is $\mathcal{L}(\mathbf{x}_{\text{static}}) = \|\mathbf{x}_{\text{static}} - \mathbf{X}_{\text{target}}\|_2^2$ which measures the point-wise Euclidean distance between the static shape and the target geometry $\mathcal{M}_{\text{target}} = (\mathbf{X}_{\text{target}}, \mathbf{T})$. In the second scenario, the loss function is $\mathcal{L}(\mathbf{x}_{\text{static}}) = \|\text{proj}_z(\mathcal{C}(\mathbf{x}_{\text{static}})) - \hat{\mathcal{C}}\|$, where $\mathcal{C}(\cdot)$ computes the center of mass of $\mathcal{M}_{\text{static}}$, $\text{proj}_z(\cdot)$ denotes the projection of the center onto the $z$-plane in world coordinates, and $\hat{\mathcal{C}}$ represents the target position for the center of mass to guarantee stability. Minimization of this function ensures the structural stability of $\mathcal{M}_{\text{static}}$.

It is crucial to highlight that the variables $\mathbf{X}_{\text{rest}}$ and $\mathbf{x}_{\text{static}}$ are tightly coupled through a hard constraint in our problem formulation. This constraint, which ensures that the object remains static equilibrium, is essential to achieving physical compatibility. Enforcing this configuration guarantees that the 3D physical object conforms strictly to external forces such as gravity, thereby ensuring the system adheres to the inherent physical constraints.

## 3.2 Parameterization of Rest-shape Geometry

To solve the optimization problem defined Eq. 2, one might consider a straightforward approach by directly treating $\mathbf{X}_{\text{rest}}$ as the optimization variable. However, this brings challenges in maintaining the physical validity of the rest-shape geometry, i.e., there shall be no inversions or inside-out elements.

This non-inversion requirement is typically enforced through nonlinear inequality constraints [11, 36], leading to intractable optimization. Drawing inspiration from natural modeling processes [42], we propose a parameterization of $\mathbf{X}_{\text{rest}}$ by treating it as the result of plastic deformation applied to an initial configuration. A *plastic deformation* can transform objects without the volume preservation constraint [1]. Specifically, we denote the initial configuration of the rest-shape geometry as $\mathcal{M}_{\text{init}} = (\mathbf{X}_{\text{init}}, \mathbf{T})$. $\mathbf{X}_{\text{rest}}$ is implicitly parameterized by the plastic deformation field $\mathbf{F}_{\mathbf{p}}$ as

$$\mathbf{X}_{\text{rest}} := \phi(\mathbf{F}_{\mathbf{p}}; \mathbf{X}_{\text{init}}), \quad \text{with} \quad \mathbf{f}_{\text{int}}(\mathbf{X}_{\text{rest}}, \mathbf{X}_{\text{init}}; \Theta) = \mathbf{0}. \tag{3}$$

Intuitively, this equation suggests that $\mathbf{X}_{\text{rest}}$ results from applying plastic strain field $\mathbf{F}_{\mathbf{p}}$ to $\mathbf{X}_{\text{init}}$ without any external forces. The plastic strain field $\mathbf{F}_{\mathbf{p}}$ is the collection of transformations, with each transformation is an $\mathbb{R}^{3 \times 3}$ matrix applied to each material point. Throughout this paper, we also represent plastic deformation in its vector form as $\mathbf{F}_{\mathbf{p}} \in \mathbb{R}^{9Z}$, which corresponds to the flattened vector form of the $\mathbb{R}^{3 \times 3}$ transformation collection. For a detailed explanation of the computation of $\mathbf{X}_{\text{rest}}$ from $\mathbf{F}_{\mathbf{p}}$ and its integration into the static equilibrium, we refer the reader to Appendix B.

There are several benefits using $\mathbf{F}_{\mathbf{p}}$ for parameterizing rest-shape geometry: It exhibits invariance to translation, which ensures that the spatial positioning of $\mathbf{X}_{\text{init}}$ does not affect the deformation outcomes. Moreover, the non-inversion requirement can be efficiently satisfied by constraining the singular values of $\mathbf{F}_{\mathbf{p}}$, thereby avoiding the need for complicated inequality constraints. Appendix B provides a comprehensive analysis of these advantages.

By substituing Eq. 3, we reformulate the optimization problem Eq. 2 as follows:

$$\min_{\mathbf{F}_{\mathbf{p}}, \mathbf{x}_{\text{static}}} \quad \mathcal{J}(\mathbf{F}_{\mathbf{p}}, \mathbf{x}_{\text{static}}) = \mathcal{L}(\mathbf{x}_{\text{static}}) + \mathcal{L}_{\text{reg}}(\mathbf{F}_{\mathbf{p}})$$
$$\text{s.t.} \quad \mathbf{f}_{\text{int}}(\mathbf{x}_{\text{static}}, \phi(\mathbf{F}_{\mathbf{p}}; \mathbf{X}_{\text{init}}); \Theta) = \mathbf{f}_{\text{ext}}(\mathbf{x}_{\text{static}}). \tag{4}$$

Here, the optimization variables are $\mathbf{F}_{\mathbf{p}}$, where the initial geometry configuration $\mathbf{X}_{\text{init}}$ is treated as a constant. The regularization term $\mathcal{L}_{\text{reg}}(\mathbf{F}_{\mathbf{p}})$ is defined as the smoothness of plastic deformation using bi-harmonic energy [5], represented as $\mathcal{L}_{\text{reg}}(\mathbf{F}_{\mathbf{p}}) = \|\mathbf{L}\mathbf{F}_{\mathbf{p}}\|_2^2$, where $\mathbf{L} \in \mathbb{R}^{9Z \times 9Z}$ denotes the graph Laplacian matrix, encapsulating the connectivity of the volumetric mesh elements.

### 3.3 Implicit Differentiation-based Optimization

Solving the optimization problem in Eq. 4 is non-trivial due to its nonlinear objective and the nonlinear hard constraint. A straightforward approach is incorporating the constraint directly into the objective as an additional loss term; however, this method may lead to imperfect satisfaction of the constraint, which undermines the fundamental goal of ensuring physical compatibility.

We resort to implicit differentiation, a technique used in sensitivity analysis [6], to compute the gradient of the objective function $\mathcal{J}$ with respect to the variable $\mathbf{F}_{\mathbf{p}}$. This approach effectively reduces the dimensionality of the optimization variables since we only need to calculate the gradient with respect to $\mathbf{F}_{\mathbf{p}}$ and also ensures that the gradient direction takes into account the hard constraint. Specifically, the gradient is computed as follows:

$$\frac{\partial \mathcal{J}}{\partial \mathbf{F}_{\mathbf{p}}} = -\left(\frac{\partial \mathcal{L}}{\partial \mathbf{x}_{\text{static}}}\right) \left[\frac{\partial \mathbf{f}_{\text{net}}}{\partial \mathbf{x}_{\text{static}}}\right]^{-1} \frac{\partial \mathbf{f}_{\text{net}}}{\partial \mathbf{F}_{\mathbf{p}}} + \frac{\partial \mathcal{L}_{\text{reg}}}{\partial \mathbf{F}_{\mathbf{p}}}, \tag{5}$$

where $\mathbf{f}_{\text{net}} = \mathbf{f}_{\text{int}} - \mathbf{f}_{\text{ext}}$ represents the net forces. A comprehensive derivation of this gradient formula is provided in Appendix C. By utilizing this gradient, the optimization can be solved using standard optimization tools, such as the Adam optimizer [17]. This facilitates the integration of our method into existing single-view reconstruction pipelines.

### 3.4 Implementation Details

Given an input image, we initially utilize off-the-shelf single-view reconstruction models to obtain the 3D object's target geometry, ensuring alignment with the input image. The output of these reconstruction models varies depending on the geometric representation used. For instance, methods employing tetrahedral representations, such as TetSphere [11], yields volumetric meshes that can be directly used as $\mathcal{M}_{\text{target}}$. Conversely, methods that output surface meshes [44] or point clouds [40], which are often non-volumetric and typically non-manifold, require additional processing steps to be suitable for our computational pipeline. We use TetWild [15], a robust tetrahedral meshing algorithm,

to convert these unstructured outputs into high-quality tetrahedral meshes, resulting in volumetric mesh $\mathcal{M}_{\text{target}}$. For initiating the optimization process, we set $\mathcal{M}_{\text{init}} = \mathcal{M}_{\text{target}}$, assuming that $\mathcal{M}_{\text{target}}$ is a reasonably good initial approximation for the optimization. Note that $\mathcal{M}_{\text{init}}$ is not strictly confined to $\mathcal{M}_{\text{target}}$; any volumetric mesh could potentially serve as the initial approximation, given the flexibility of $\mathbf{F_p}$ to accommodate spatially varying deformations.

For the material constitutive model, we use isotropic Neo-Hookean material as detailed in [35]. The mechanical properties $\Theta$, including Young's modulus $E$, Poisson's ratio $\nu$, and mass density $\rho$, are set by users. These values can be specified directly through numerical input or chosen from a collection of pre-established material options, such as plastic or rubber. We consider gravity and fixed attachment forces as options for external forces. Gravity is always included to reflect its omnipresence in the real world. The use of fixed attachment forces depends on the specific needs of the application, for instance, anchoring an object at a designated site. Detailed formulations for both force types are provided in Appendix F.

## 4 Evaluation

In this section, we present evidence that our approach enhances the physical compatibility of 3D objects produced using state-of-the-art single-view reconstruction techniques. We conduct a series of quantitative evaluations using five metrics (Sec. 4.1) to compare the physical compatibility of shapes optimized by our framework against those produced by existing methods without our method (Sec. 4.2). We also provide qualitative comparisons to demonstrate to the effectiveness of our approach (Sec. 4.3). Furthermore, we explore the practical applications of our method by illustrating how it enables the reconstruction of diverse 3D shapes with different material properties from the same single image, and by demonstrating that our optimized shapes are readily adaptable for dynamic simulations and fabrication (Sec. 4.4).

### 4.1 Baselines and Evaluation Protocol

Existing metrics for evaluating single-view reconstruction methods primarily focus on the visual appearance of the objects. Measures such as PSNR and SSIM are used to assess image fidelity, while chamfer distance and volume IoU evaluate geometric quality. However, these metrics do not consider the underlying physics principles that govern the behavior of 3D objects. Consequently, they are insufficient for evaluating the physical compatibility of reconstructed shapes, a crucial aspect for applications requiring accurate physical interactions and structural stability.

**Metrics.** To address this oversight, we draw inspiration from the field of finite element analysis [2] and introduce five novel metrics specifically designed to assess the physical compatibility of 3D models comprehensively. These metrics are tailored to ensure a more thorough evaluation of method performance in real-world scenarios with rich physics:

- **Number of Connected Components (#CC.)** evaluates the structural integrity of the object. Physical objects should not have floating or disconnected structures, ideally consisting of one single connected component.
- **Mean Stress** calculates the average von Mises stress [28] across all tetrahedra of all objects. It measures the extent of physical deformation. Under the same external interactions, higher mean stress indicates a greater likelihood of fracture and the existence of unrealistic thin structures.
- **Percentage of Standability (Standable.)** assesses whether the object can maintain stability under gravity, remaining upright without toppling. A standable object is one that effectively supports itself against gravitational forces.
- **Matching loss (Img. Loss)** calculates the $l_1$ difference between the rendered image of the object after applying gravity and the input target image, quantifying the deviation of the physical object from the desired shape due to physical influences.
- **Fracture Rate** measures the number of tetrahedral elements that exceed a predefined stress threshold, potentially leading to fractures. The resilience of a method against physical stresses is quantified using a degradation curve, with more physically reliable methods exhibiting a smaller area under the curve for the fracture rate.

**Baselines.** We consider five single-view reconstruction baselines in our evaluation, each associated with a distinct geometry representation: Wonder3D [23] with NeuS, LGM [40] with Gaussian

Table 1: Quantitative results on four metrics evaluating physical compatibility. We apply our pipeline to five single-image reconstruction techniques and assess our metrics on both the initial shapes from these methods (Baseline) and the optimized shapes from the integration of our framework with each baseline (Ours). Our method demonstrates quantitative improvements in mean stress, stability rate, and image fidelity across all benchmarks. Among all methods, TetSphere integrated with our framework achieves superior performance across all evaluation metrics. This can be attributed to the explicit volumetric representation used in TetSphere. The mean and standard deviation are calculated across all examples for each method. A higher deviation in Mean Stress suggests a larger variance in structural thickness and curvature, while a higher deviation in Img. Loss indicates a larger variance in static shape deformation.

| Method | | Init. Geo. | #CC. ↓ | Mean Stress ↓ (kPa) | Standable. ↑ (%) | Img. Loss ↓ |
|---|---|---|---|---|---|---|
| **Wonder3D** | Baseline | NeuS | 2.54 ± 2.64 | 10.68 ± 17.47 | 6.9 | 0.073 ± 0.063 |
| | Ours | | | 0.45 ± 0.96 | 72.4 | 0.069 ± 0.048 |
| **LGM** | Baseline | Gaussian | 2.67 ± 2.13 | 1.14 ± 2.03 | 20.3 | 0.121 ± 0.091 |
| | Ours | splatting | | 1.01 ± 1.34 | 85.5 | 0.116 ± 0.065 |
| **MeshLRM** | Baseline | surface | 1.55± 2.13 | 0.54 ± 1.41 | 29.6 | 0.065 ± 0.042 |
| | Ours | mesh | | 0.38 ± 1.05 | 74.5 | 0.064 ± 0.042 |
| **TripoSR** | Baseline | NeRF | 1.43 ± 1.12 | 0.29 ± 1.28 | 24.2 | 0.066 ± 0.047 |
| | Ours | | | 0.22 ± 0.94 | 80.6 | 0.059 ± 0.039 |
| **TetSphere** | Baseline | tet-sphere | **1.00 ± 0.00** | 0.22 ± 0.51 | 32.8 | 0.061 ± 0.045 |
| | Ours | | | **0.19 ± 0.78** | **92.2** | **0.057 ± 0.040** |

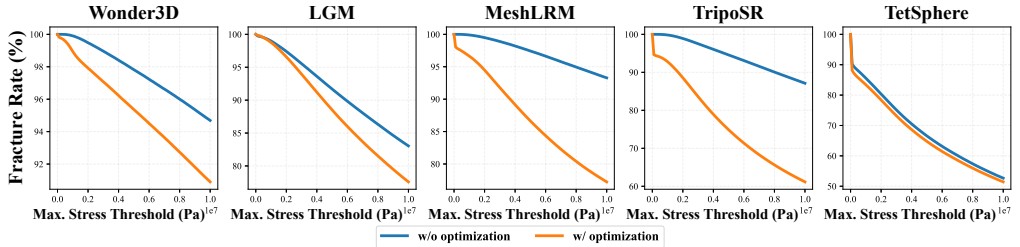

Figure 3: Quantitative results on fracture rate. We plot the relationship between the fracture rate and the maximum stress threshold across five single-image reconstruction methods. The shapes optimized with our framework exhibit a consistently lower fracture rate compared to those shapes obtained without our pipeline. MeshLRM and TripoSR feature prevalent thin structures in their reconstructed shapes, whereas our approach significantly reduces the fracture rate in both cases.

splatting, MeshLRM [44] with surface mesh, TripoSR [41] with NeRF triplane, and TetSphere [11] with tetrahedral spheres. For the baseline results, we used the publicly available inference code to reconstruct the 3D objects.[2] To demonstrate the versatility of our method, we integrated our physical compatibility optimization framework with all five baseline models and reported the results to ensure a fair comparison. The implementation details of our framework are provided in Appendix D.

**Evaluation Datasets.** The evaluation dataset was sourced from Objaverse [9]. We initially randomly selected approximately 200 shapes from the categories of plants, animals, and characters – categories that demand greater physical compatibility. Single-view images were rendered using the publicly released code by the authors of Objaverse[3]. Subsequently, these images were used to reconstruct 3D objects using the baseline methods mentioned earlier. We filtered out shapes of extremely poor quality, specifically those with more than 8 connected components. This process resulted in a final set of 100 shapes for detailed evaluation.

Despite these shapes being a part of the training data for most baseline methods, our evaluation focuses on assessing the physical compatibility – a factor overlooked by these methods. The results obtained from this dataset provide valuable insights and observations on the physical compatibility of each method, demonstrating the practical effectiveness of our approach.

---

[2]For MeshLRM, since the pre-trained model is not publicly available yet, we obtained the reconstructed shapes directly from the authors for use in our study.

[3]https://github.com/allenai/objaverse-rendering

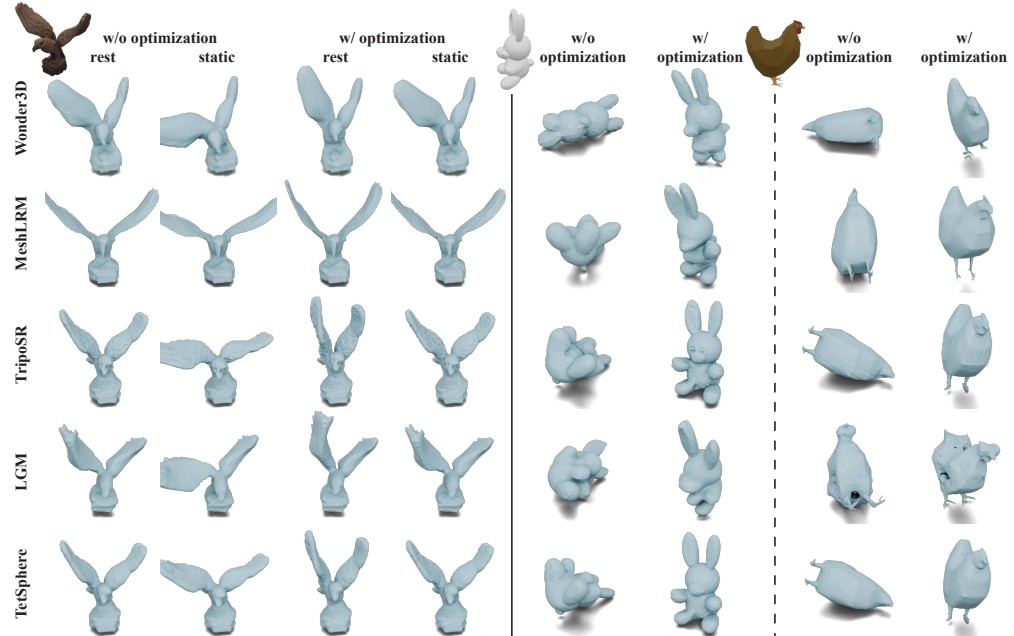

Figure 4: Qualitative results on physical compatibility optimization. Left: Rest shapes optimized using our approach result in static shapes that closely match the input images when subjected to gravity. In contrast, shapes without the optimization fail to replicate the geometry in the input image. Right: our optimization process ensures that the optimized shapes are capable of supporting themselves, whereas the baseline methods fail to achieve this stability.

## 4.2 Quantitative Results

Table 1 shows the quantitative results for four out of five metrics evaluated for both baselines and those integrated with our physical compatibility optimization. Fig. 3 shows the curve of fracture rate.

Our quantitative analysis yields several observations: 1) The underlying geometry representation significantly impacts the structural integrity of reconstructed shapes, as evidenced by the number of connected components (#CC.). LGM, using a point cloud representation, exhibits the poorest structural integrity, often resulting in floating structures due to its inability to differentiate the interior from the exterior of a 3D object. In contrast, TetSphere, with its volumetric representation, maintains the most integral structure. 2) Both MeshLRM and TripoSR generally produce more physically stable 3D objects, as indicated by Mean Stress and Standability (Standable.) metrics. However, they tend to diverge under gravity, as shown by the Matching Loss metric (Img. Loss), compared to TetSphere. 3) Notably, our method consistently enhances the physical compatibility performance across all baselines. The improvement is particularly significant for Wonder3D and MeshLRM. Wonder3D typically generates multi-view images before reconstructing the 3D shape, which can lead to thin structures due to inconsistencies across the views. Similarly, MeshLRM's reliance on surface mesh could often result in thin structures. Our method strengthens the physical robustness for both cases. 4) Our method also enhances the structure robustness to fracture, as demonstrated in Fig. 3. It notably improves the performance of both MeshLRM and TripoSR in reducing fracture rates.

## 4.3 Qualitative Results

Fig. 4 and more qualitative results in Appendix 4 illustrate the effectiveness of our physical compatibility optimization. Without optimization, the static shapes behave undesirably under general physical principles: they either sag excessively under gravity, diverging from the geometry depicted in the input image, or fail to remain upright, toppling over. Our optimization method incorporates physical principles to ensure that the optimized rest shapes are self-supporting and stable, and match the input images under static equilibrium.

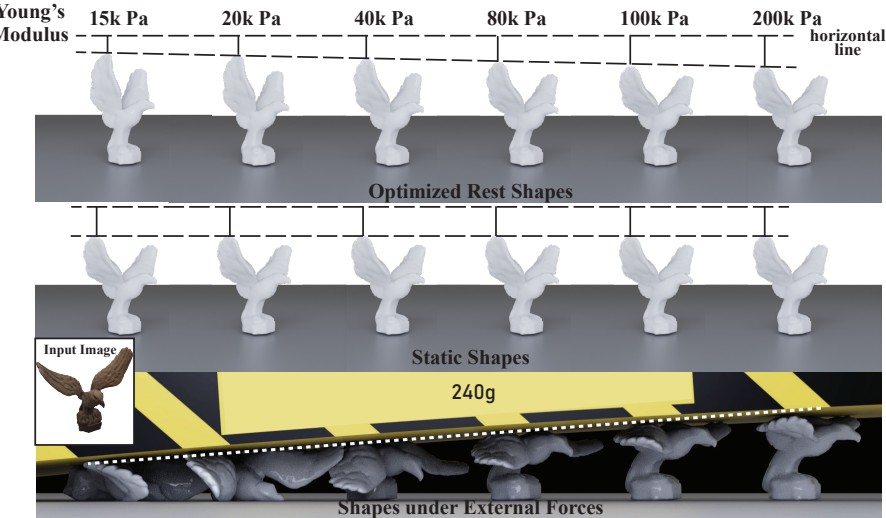

Figure 5: Ablation study on Young's modulus. By changing the material properties, our method can produce various rest-shape geometries (top), which all result in the same static shapes that match the input image (middle). Although these static shapes appear identical under static equilibrium, they exhibit different deformation when subjected to the same compression forces exerted by the yellow block, attributable to the differences in their material properties (bottom).

## 4.4 Analysis

**Ablation study on Young's Modulus.**   We investigate the influence of predefined mechanical material properties, particularly Young's modulus, on the optimized rest shapes and their physical behaviors. Using the same input image, we obtained six optimized rest shapes with varying Young's modulus values within our framework with TetSphere. As shown in Fig. 5, although the optimized rest-shape geometries vary, they all deform to the same static geometry under the influence of gravity, matching the input image. Moreover, the physical responses to identical external forces, such as compression by a box, differ due to the variations in material properties. These results highlight how the explicit decomposition of physical attributes in our framework expands the controllability of object modeling, allowing for diverse physical behaviors under uniform external forces.

**Application to dynamic simulation.**   The immediate output of our method is a simulation-ready rest-shape geometry, which can be seamlessly integrated into a simulation pipeline to produce complex dynamics and motions. Fig. 6 (left) and the accompanying video in the Supplementary Material illustrate three plants modeled using our framework, demonstrating their behavior under gravity and complex interactions. Implementation details of this simulation are provided in Appendix F. These examples underscore the practical utility of our method for generating physically realistic dynamics and simulations.

**Application to fabrication.**   We further evaluate our method in real world by fabricating three shapes using 3D printing, both with and without optimization. The results, shown in Fig. 6 (right), with detailed implementation procedures available in Appendix E, demonstrate that the 3D printed shapes align with our computational results. These real-world experiments demonstrate the practical effectiveness and validate the physical realism of the objects produced by our method.

## 5   Conclusion

In this work, we introduced physical compatibility optimization for reconstructing a physical object from a single image. Our method decomposes three orthogonal attributes governing physical behavior: mechanical properties, external forces, and rest-shape geometry. Unlike existing methods that often ignore one or more dimensions, our framework holistically considers all three factors, allowing for diverse rest-shape geometries from the same input image by varying object stiffness and external

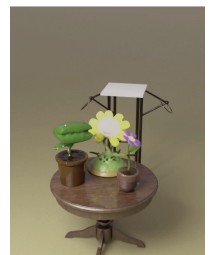
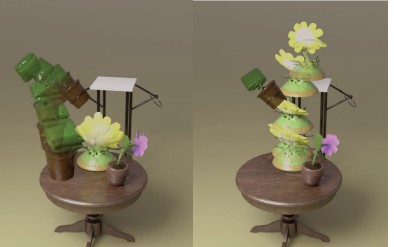
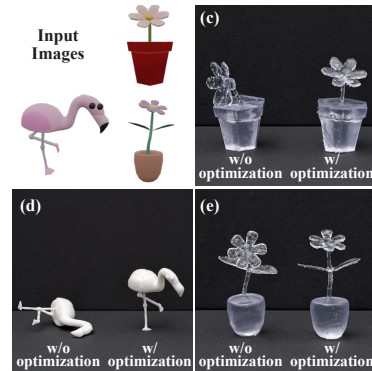

(a) Static shapes under gravity    (b) Dynamic simulation

Figure 6: Applications of physically compatible objects. Left: Our optimized physical objects is simulation-ready and can be seamlessly integrated into dynamic simulation pipeline to produce complex dynamics and motions. Right: Real-world validation using 3D printing shows that shapes optimized using our method closely replicate the input images, demonstrating the practical effectiveness of our method in manufacturing.

forces. We formulate physical compatibility optimization as a constrained optimization problem by integrating static equilibrium as a hard constraint. Our approach produces physical objects that match the geometry depicted in the input image under external forces and remain stable under gravity. Both quantitative and qualitative evaluations demonstrated improvements in physical compatibility over existing baselines. Our method's versatility is evident through its integration with various single-view reconstruction methods and its practical applications in dynamic simulations and 3D printing.

**Limitations and Future Work**   One limitation of our framework is its reliance on predefined material properties and external forces as inputs. Although this provides controllability of the final optimized rest-shape geometry, automating the extraction of these parameters from a single image presents a potential avenue for future work. Moreover, our method relies on the use of a tetrahedral mesh, which is derived by tetrahedralizing the output geometry produced by baseline methods. A natural extension of our work is the development of a differentiable converter that can transform any geometric representation into a tetrahedral mesh. This would enable future research where our physical compatibility optimization could be integrated into a pre-trained large reconstruction model, which could then be fine-tuned to directly produce physically compatible 3D objects. Lastly, our current methodology focuses solely on physical objects in a state of static equilibrium. Exploring the reconstruction of 3D objects undergoing dynamics captured from video is an intriguing prospect for future research.

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

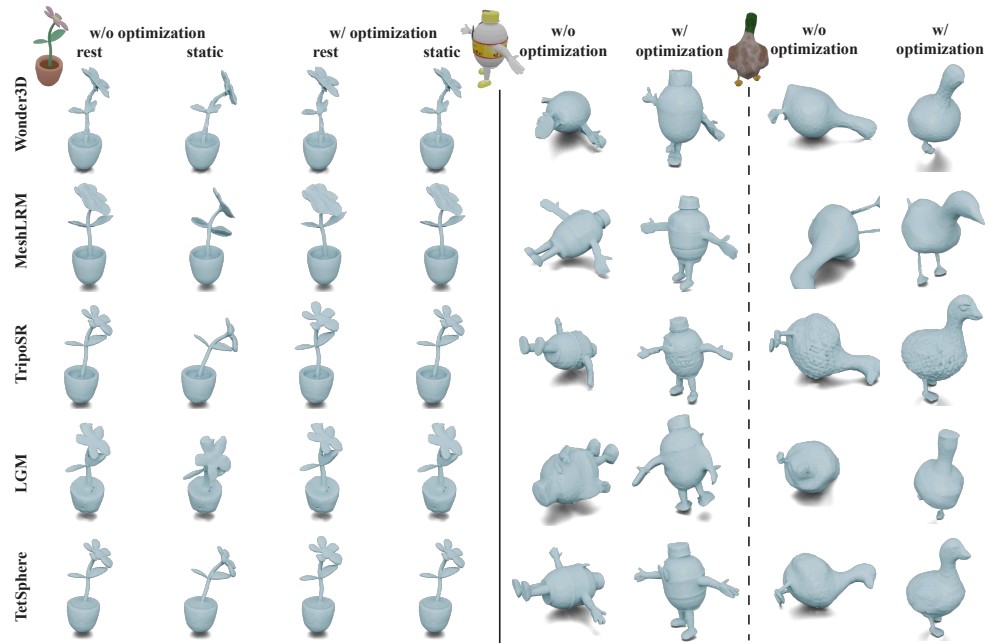

Figure 7: Additional qualitative results of physical compatibility optimization (part 1/2).

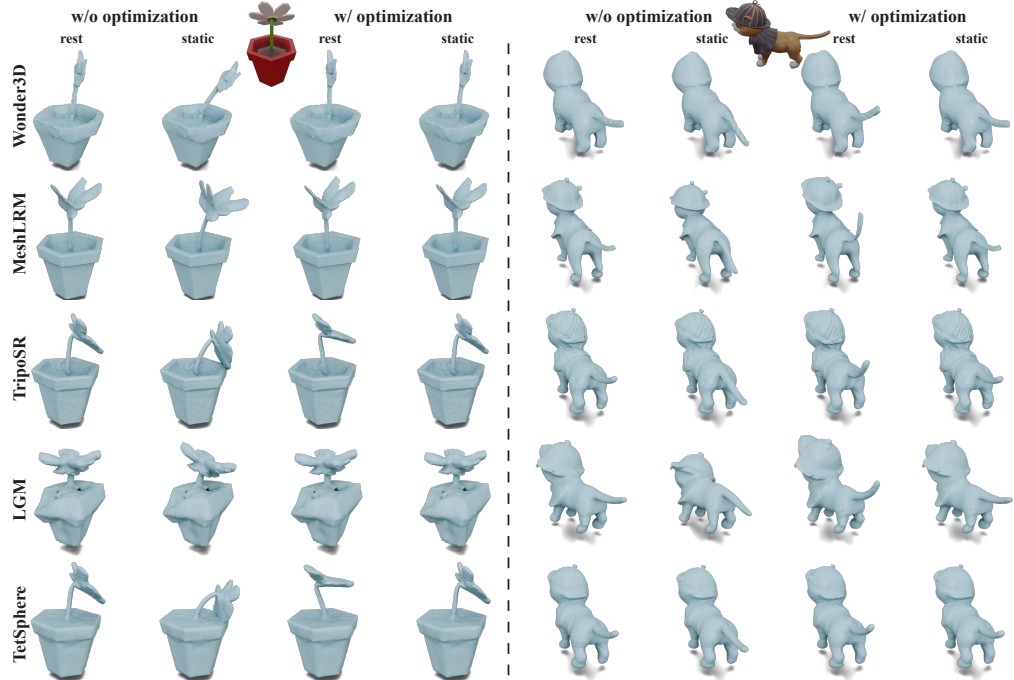

Figure 8: Additional qualitative results of physical compatibility optimization (part 2/2).

## A Additional Qualitative Results

Figure 7 and 8 show additional results of our physical compatibility optimization.

## B Plastic Strain Field $\mathbf{F_p}$

To enhance the understanding of our framework without compromising generalizability, let us consider $\mathcal{M}_{\text{init}}$ to be a tetrahedral mesh composed of a single element and four vertices. When

subject to static equilibrium influenced by gravity, the object adheres to the equation:

$$\mathbf{f}_{\text{int}}(\mathbf{x}, \phi(\mathbf{F_p}; \mathbf{X}_{\text{init}}); \Theta) = \mathbf{Mg}, \tag{6}$$

where $\mathbf{f}_{\text{int}}(\cdot, \cdot)$ denotes the elastic force (internal force), $\mathbf{M}$ is the mass matrix, and $\mathbf{g}$ denotes the gravity acceleration. To compute this force, we first consider the elastic energy $\mathcal{E}$. The definition of elastic energy unfolds as follows:

$$\mathcal{E}(\mathbf{F_e}, \mathbf{F_p}; \Theta) = V(\mathbf{F_p})\Phi(\mathbf{F_e}; \Theta),$$
$$V(\mathbf{F_p}) = V_{\text{init}}\det(\mathbf{F_p}),$$
$$\mathbf{F_e} = \mathbf{F}\mathbf{F_p}^{-1},$$
$$\mathbf{F} = \partial\mathbf{x}/\partial\mathbf{X}_{\text{init}},$$

where $V(\mathbf{F_p})$ represents the volume of the element under plastic strain, $V_{\text{init}}$ is the initial volume of the element, $\mathbf{F_e}$ denotes the elastic deformation gradient, $\mathbf{F}$ represent the total deformation gradient, and $\Phi(\cdot; \Theta)$ the elastic energy density function. This deformation gradient $\mathbf{F}$ is computed through standard methodology [34].

Consequently, the derivation of the elastic force encapsulates the computation of the first-order partial derivative of the elastic energy with respect to the vertex positions:

$$\mathbf{f}_{\text{int}}(\mathbf{x}, \phi(\mathbf{F_p}; \mathbf{X}_{\text{init}}); \Theta) := \frac{\partial\mathcal{E}(\mathbf{F_e}(\mathbf{x}), \mathbf{F_p}; \Theta)}{\partial\mathbf{x}}$$
$$= V(\mathbf{F_p})\frac{\partial\Phi}{\partial\mathbf{F_e}} : \frac{\partial\mathbf{F}}{\partial\mathbf{x}}\mathbf{F_p}^{-1}.$$

Notably, given the linear dependence of $\mathbf{F}$ on $\mathbf{x}$, $\frac{\partial\mathbf{F}}{\partial\mathbf{x}}$ remains constant.

Given $\mathbf{F_p}$ and $\mathbf{X}_{\text{init}}$ as inputs, the solution to Eq. 6 is the static shape, $\mathbf{x} = \mathbf{x}_{\text{static}}$. Likewise, to calculate $\mathbf{X}_{\text{rest}}$ from $\mathbf{F_p}$ and $\mathbf{X}_{\text{init}}$ in Eq. 3, we solve a similar equation with zero external force.

$$\mathbf{f}_{\text{int}}(\mathbf{x}, \phi(\mathbf{F_p}; \mathbf{X}_{\text{init}}); \Theta) = \mathbf{0},$$

where the solution to this equation is $\mathbf{X}_{\text{rest}}$.

Considering the elastic energy, the translation of $\mathbf{X}_{\text{init}}$ does not alter the deformation gradient $\mathbf{F}$. Consequently, $\mathbf{F_p}$ remain unaffected and exhibit translation invariance. In terms of the elastic force, it maintains translation invariance as well, since $\mathbf{F}$ is not affected by any shift in $\mathbf{X}_{\text{init}}$.

Finally, by using isotropic materials, our approach enables a further reduction in the DOFs of $\mathbf{F_p}$. Let us denote $\mathbf{F_p}$ as $\mathbf{F_p} = \mathbf{RS}$. The elastic deformation gradient is then derived as $\mathbf{F_e} = \mathbf{F}(\mathbf{RS})^{-1} = \mathbf{F}\mathbf{S}^{-1}\mathbf{R}^{-1}$. Given the invariance property $\Phi(\mathbf{F_e}; \theta) = \Phi(\mathbf{F_e}\mathbf{R}; \theta)$, which constantly holds for isotropic materials, the rotation component $\mathbf{R}$ becomes redundant and can be excluded from the formulation. This simplification implies that the only requirement for $\mathbf{F_p}$ is to be a symmetric matrix. During the optimization process, this property facilitates the prevention of the inversion: In order to ensure that $\det(\mathbf{F_p}) > 0$, we can simply adjust the eigenvalues of $\mathbf{F_p}$ to make they remain positive. This adjustment is crucial for the rest mesh $\mathbf{X}_{\text{rest}}$ to maintain in the non-inverted state.

## C  Computation of Gradient

By differentiating the constraint in Eq. 4 with respect to $\mathbf{F_p}$, we obtain

$$\frac{\partial\mathbf{f}_{\text{net}}}{\partial\mathbf{F_p}} + \frac{\partial\mathbf{f}_{\text{net}}}{\partial\mathbf{x}_{\text{static}}}\frac{\partial\mathbf{x}_{\text{static}}}{\partial\mathbf{F_p}} = 0. \tag{7}$$

Then, we have

$$\frac{\partial\mathbf{x}_{\text{static}}}{\partial\mathbf{F_p}} = -[\frac{\partial\mathbf{f}_{\text{net}}}{\partial\mathbf{x}_{\text{static}}}]^{-1}\frac{\partial\mathbf{f}_{\text{net}}}{\partial\mathbf{F_p}}. \tag{8}$$

Substituting the result into the objective in Eq. 4, we get

$$\frac{\partial\mathcal{J}}{\partial\mathbf{F_p}} = \frac{\partial\mathcal{L}}{\partial\mathbf{F_p}} + \frac{\partial\mathcal{L}_{\text{reg}}}{\partial\mathbf{F_p}}$$
$$= \frac{\partial\mathcal{L}}{\partial\mathbf{x}_{\text{static}}}\frac{\partial\mathbf{x}_{\text{static}}}{\partial\mathbf{F_p}} + \frac{\partial\mathcal{L}_{\text{reg}}}{\partial\mathbf{F_p}}$$
$$= -\frac{\partial\mathcal{L}}{\partial\mathbf{x}_{\text{static}}}[\frac{\partial\mathbf{f}_{\text{net}}}{\partial\mathbf{x}_{\text{static}}}]^{-1}\frac{\partial\mathbf{f}_{\text{net}}}{\partial\mathbf{F_p}} + \frac{\partial\mathcal{L}_{\text{reg}}}{\partial\mathbf{F_p}}, \tag{9}$$

which is the gradient with respect to $\mathbf{F_P}$. In practice, $\frac{\partial \mathbf{f}_{\text{net}}}{\partial \mathbf{x}_{\text{static}}}$ and $\frac{\partial \mathbf{f}_{\text{net}}}{\partial \mathbf{F_P}}$ are stored as sparse matrices and computed based on [42]. Considering about the performance, we first compute $\frac{\partial \mathcal{L}}{\partial \mathbf{x}_{\text{static}}} \left[ \frac{\partial \mathbf{f}_{\text{net}}}{\partial \mathbf{x}_{\text{static}}} \right]^{-1}$ using sparse linear solver. This results in a dense vector with size $3N$. We then multiply it with $\frac{\partial \mathbf{f}_{\text{net}}}{\partial \mathbf{F_P}}$.

## D  Implementation Details of Evaluation

To evaluate the physical compatibility of baseline methods, which often produce shapes comprising multiple connected components, we first extract the largest connected component from each mesh. All meshes are then normalized to the unit cube. Notably, the reconstructed shapes from TripoSR and Wonder3D are not axis-aligned; thus, we manually rotate these shapes to ensure the head points towards the $z$-axis in the world coordinate space. For integrating our physical compatibility framework, We use two sets of Young's modulus, $E = 5 \times 10^4 \text{Pa}$ and $E = 5 \times 10^5 \text{Pa}$, which are selected based on whether the shape would become overly soft, potentially leading to static equilibrium failure due to excessive stress causing numerical bounds to be exceeded. Poisson's ratio $\nu = 0.45$ and mass density $\rho = 1000 \text{kg/m}^3$ are consistent across all meshes. Evaluation metrics require solving for static equilibrium Eq. 1. We employ the Newton-Raphson solver with line search, setting the maximum number of iterations to be 200. For optimizing Eq. 4, we use gradient descent and allow up to 1000 iterations. Our experiments run on a desktop PC with an AMD Ryzen 9 5950X 16-core CPU and 64GB RAM. The average runtime for this optimization process is approximately 80 seconds.

## E  Implementation Details of 3D Printing

The selected model shapes were 3D printed using stereolithography (Form3; Formlabs, 100 $\mu$m layer thickness) to create the flexible designs (using Flexible 80A, tensile modulus <3 MPa, 100% strain to failure) and rigid designs (using White Resin V4; tensile modulus 1.6 GPa), both without post-curing. The flexible flowers are 55 and 65 mm in height and the rigid goose is 50 mm in length. Shapes with and without optimization were printed with similar support structures designed to preserve delicate features.

## F  Dynamic Simulation of Deformable Objects

We model each solid deformable object using FEM with hyperelastic materials for dynamic simulation. Then, we solve the standard partial differential equation (PDE) for dynamic FEM simulation:

$$M\ddot{x} + D(x)\dot{x} + f_{\text{elastic}}(x) + f_{\text{attachment}}(x) + f_{\text{contact}}(x) = Mg, \qquad (10)$$

where $x$ represents the node positions within the finite element meshes – we use tetrahedral meshes – of the objects, $M$ denotes the mass matrix, $D$ is the Rayleigh damping matrix, $f_{\text{elastic}}(\cdot)$ is the hyperelastic forces, $f_{\text{attachment}}(x)$ is the attachment forces that constrain the objects to a specific location, and $f_{\text{contact}}(\cdot)$ denotes the contact forces between surfaces. We employ the implicit backward Euler method for time discretization, transforming the PDE into:

$$A^n x^{n+1} + b^n + f_{\text{elastic}}(x^{n+1}) + f_{\text{attachment}}(x^{n+1}) + f_{\text{contact}}(x^{n+1}) = 0, \qquad (11)$$

where $x^{n+1}$ is the position vector at timestep $(n+1)$, $A^n$ and $b^n$ is a constant matrix and vector, respectively, derived from values at timestep $n$, Finally, we solve this nonlinear equation using Newton's method at each timestep.

The hyperelastic material selected for the deformable objects is the same as the one used for the rest shape optimization [35] in Sec. 3. Attachment forces are modeled as spring forces $f_{\text{attachment}}(x) = k_a(Sx - \bar{x}(t))$, where $k_a$ is the stiffness of the spring, the selection matrix $S$ selects the attached vertices, and $\bar{x}(t)$ denotes the target attachment locations at time $t$. Contact forces are generated from penalizing any vertex penetration into the contact surface, expressed as $f = k_c d$, where $k_c$ represents the contact stiffness and $d$ denotes the penetration depth, with $d = 0$ in the absence of contact. This gives the normal contact forces. Friction forces are computed following the methods outlined in [18]. Then, the total contact force $f_{\text{contact}}$ is the sum of normal contact forces and friction forces.

For the dynamic simulation in Figure 7, the attachment of each plant is defined as the bottom part of each pot. We keyframe-animate the trajectory of attachment vertices $\bar{x}(t)$. Gravity is enabled

throughout the entire simulation. At the end of the sequence, we apply wind forces to the plants, computed using 4D Perlin Noise [30].

## G   Broader Impacts

Our research presents a computational framework for reconstructing physical objects from single images. This advancement holds significant potential for various applications, including dynamic simulations, 3D printing, virtual reality, and industrial design. By ensuring that the reconstructed objects adhere to real-world physical laws, our method can enhance the realism and functionality of virtual environments, improve the precision of 3D printed objects, and contribute to the development of more reliable industrial designs.

There are mainly two potential negative societal impacts: Improved 3D reconstruction capabilities could potentially be misused to create highly realistic fake objects or environments for disinformation purposes. This could include generating deceptive media content that appears authentic. As the framework automates the reconstruction process, there is a potential risk of it being used in automated systems without sufficient oversight, potentially leading to unintended and harmful outcomes due to errors or misuse. Developing systems to monitor the use of the technology and ensure accountability for its applications, as well as providing comprehensive guidelines and training for users to promote ethical use and awareness of potential misuse, will address these potential negative impacts.

