# OpenReview forum: "Physically Compatible 3D Object Modeling from a Single Image"
_NeurIPS.cc/2024/Conference — NeurIPS 2024 spotlight_

### Official Review · Reviewer_XLVo · 2024-06-15

**Soundness:** 3
**Presentation:** 4
**Contribution:** 4
**Rating:** 7
**Confidence:** 4

**Summary:**

This paper presents an optimization method that produces 3D shapes from a single images that considers about mechanical properties and external forces. The shapes of the generated results in a state of static equilibrium are able to match the input image. The proposed methods can be used with other single-view image shape generation methods.

**Strengths:**

The strengths of the proposed method are:
- The proposed optimization framework is novel and versatile.
- The method supports to produce fabricatable shapes from a single image.

**Weaknesses:**

The weaknesses of the proposed method are:
- The limitation of the proposed method is not discussed in the main paper.
- The discussion with alternative approaches is not clear in the main paper.

**Questions:**

- I am curious about why the authors decided to optimize the deformation instead of optimizing the hollow of the solid geometry, similar to the method used in Make-It-Stand [Prevost et al. 2013] and the follow-up papers.
- It is unclear to me what is the limitation of the proposed method.
- It is unclear to me what is the performance overhead for the proposed method. I would encourage the authors to report the shape complexity and the corresponding computational time.
- It is unclear to me whether image loss is used during the optimization? In fig. 2, the pipeline invovles image loss but I did not see the discussion in Sec. 2 but only in evaluation part.

**Limitations:**

As mentioned above, I think the limitations should be discussed more in the main paper.

---

> ### Author Rebuttal · Authors · 2024-08-06
>
> We appreciate the valuable comments from the reviewer.
>
> **Rebuttal Outline:**
>
> 1. **[Adjustment] (W1&W2&Q2)** Move Limitations and Discussions with Alternative Approaches to Main Text.
> 2. **[Discussion] (Q1)** Alternatively Optimizing the Hollow of the Solid Geometry.
> 3. **[Clarification] (Q3)** Performance.
> 4. **[Clarification] (Q4)** Image Loss in Optimization.
>
> Please find below our detailed responses to your comments and concerns.
>
> ---
>
> **1. (W1&W2&Q2) The limitation of the proposed method and the discussion with alternative approaches is not clear in the main paper.**
>
> Thank you for pointing out the placement of our discussions on the method's limitations and comparisons with alternative approaches. These discussions are indeed present in Appendices G and H. We acknowledge the importance of making this information more accessible and are considering moving these sections into the main body of the text. We plan to compress Figure 4 and move Figure 5 to the appendix to free up enough space to accommodate these critical discussions within the page limit.
>
> ---
>
> **2. (Q1) I am curious about why the authors decided to optimize the deformation instead of optimizing the hollow of the solid geometry, similar to the method used in Make-It-Stand [Prevost et al. 2013] and the follow-up papers.**
>
> While optimizing the hollow of solid geometry is suitable for enhancing standability, it is not well-suited to address other losses such as ensuring the static shape accurately replicates the geometry depicted in the input image. Furthermore, Make-It-Stand and its follow-ups involve an extra carving stage that hollows out the interior of the 3D shape, along with a shape deformation stage. This carving process often depends on manually defined heuristics to prevent the creation of isolated parts and is computationally intensive due to its combinatorial nature. In contrast, our method using the plastic deformation on the whole shape eliminates the need for this additional and complex stage.
>
> ---
>
> **3. (Q3) It is unclear to me what is the performance overhead for the proposed method. I would encourage the authors to report the shape complexity and the corresponding computational time.**
>
> The average shape complexity in our experiments, across 100 evaluated shapes, is approximately 5,200 vertices and 14,000 tetrahedra per shape. Regarding computational performance, the average computational time required for our method on a standard PC configuration (AMD Ryzen 9 5950X 16-core CPU and 64GB RAM) is about 2.5 minutes per shape. We will include these results in the final manuscript.
>
> ---
>
> **4. (Q4) It is unclear to me whether image loss is used during the optimization? In Fig. 2, the pipeline involves image loss, but I did not see the discussion in Sec. 2, only in the evaluation part.**
>
> Thank you for raising this point. We did not utilize image loss directly during the optimization process. Instead, we assume that the initially reconstructed shape from the input image accurately represents the target geometry, as outlined in lines 161-162. The evaluation of image loss, however, is used to assess whether the final optimized shape, when subjected to external physical forces such as gravity, still corresponds closely with the original input image.
>
> ---
> Please let us know if you have any further questions. We really appreciate your time. Thank you!
>
> Best regards,
>
> Authors

---

### Official Review · Reviewer_91LB · 2024-07-12

**Soundness:** 4
**Presentation:** 4
**Contribution:** 3
**Rating:** 6
**Confidence:** 3

**Summary:**

This paper introduces the concept of physical compatibility into single-image to 3D generation. It proposes a post-optimization method for existing 3D generation pipelines, which takes the generated mesh as the target mesh and optimizes the deformation gradient to obtain a rest-shape geometry without external forces. This rest-shape geometry can align with the target mesh under physical conditions provided by the user. The proposed method significantly improves the results for different single-image to 3D pipelines and is demonstrated to be effective through a comparison of 3D printed results under real-world scenarios.

**Strengths:**

The idea of introducing physical compatibility into 3D generation is very innovative, and it is crucial for the application of current 3D generation results to real-world scenarios. The authors provide detailed derivations and explanations, and propose five reasonable evaluation metrics to assess the degree of physical compatibility. They demonstrate the effectiveness of their method in real-world scenarios through 3D printed objects.

**Weaknesses:**

This is not strickly a weakness of the proposed method. The paper mainly focuses on the optimization of a single object. However, current 3D generation pipelines can handle the generation of multiple coupled objects, such as a castle. I wonder whether this method can be applied to such senarios.

**Questions:**

The proposed method optimizes the positions of mesh vertices while maintaining the connectivity of the vertices. However, existing 3D generation models may introduce incorrect connectivity for complex or partially occluded input image. I wonder if physical compatibility approach can correct such errors in connectivity during the optimization process.

**Limitations:**

Yes.

---

> ### Author Rebuttal · Authors · 2024-08-06
>
> We are grateful for the reviewer’s insightful comments.
>
> **Rebuttal Outline:**
>
> 1. **[Discussion] (W1)** Handling the generation of multiple coupled objects.
> 2. **[Discussion] (Q1)** Correcting errors in connectivity during the optimization process.
>
> Please find below our detailed responses to your comments and concerns.
>
> ---
>
> **1. (W1) This is not strictly a weakness of the proposed method. The paper mainly focuses on the optimization of a single object. However, current 3D generation pipelines can handle the generation of multiple coupled objects, such as a castle. I wonder whether this method can be applied to such scenarios.**
>
> We appreciate your observation highlighting a promising direction for future research. Extending our method to handle multiple objects introduces the challenge of managing interactions and collisions between objects. As a potential pathway forward, we could integrate differentiable simulation techniques and collision constraints [1] into existing 3D generative models. This adaptation would allow our framework to address the dynamic interactions and physical compatibility of multiple coupled objects.
>
>
> ---
>
> **2. (Q1) Existing 3D generation models may introduce incorrect connectivity for complex or partially occluded input images. I wonder if the physical compatibility approach can correct such errors in connectivity during the optimization process.**
>
> Currently, our method maintains fixed connectivity during the physical compatibility optimization. However, a promising extension to address incorrect initial connectivity could involve incorporating adaptive remeshing or subdivision techniques during the optimization process. Specifically, while adhering to our established physical compatibility framework, we could intersperse the optimization with periodic remeshing steps. This approach would adaptively correct connectivity errors throughout the optimization process, enhancing the accuracy and robustness of the reconstructed shapes.
>
> ---
> **References**
>
> [1] Huang, Zizhou, et al. "Differentiable solver for time-dependent deformation problems with contact." ACM Transactions on Graphics 43.3 (2024): 1-30.
>
> ---
> We sincerely appreciate the comments. Please let us know if you have further questions.
>
> Best regards,
>
> Authors

---

### Official Review · Reviewer_H9KU · 2024-07-12

**Soundness:** 4
**Presentation:** 3
**Contribution:** 4
**Rating:** 5
**Confidence:** 4

**Summary:**

This paper introduces a physical compatibility optimization framework for reconstructed objects from a single image. The approach considers mechanical properties, external forces, and rest-shape geometry, integrating static equilibrium as a hard constraint. This framework improves upon existing methods by ensuring the stability and accuracy of reconstructed objects under external influences. Quantitative and qualitative evaluations show enhancements in physical compatibility.

**Strengths:**

1. Performance: The proposed method achieves state-of-the-art results. The experiments well validate the effectiveness of the proposed methods.

2. Clarity: The paper is well-written and easy to follow.

3. Technical Novelty: The main contributions of this paper are twofold: 1) They propose a physical compatibility optimization framework for 3D objects and decompose the mechanical properties, external forces, and rest-shape geometry. 2) They optimize the rest-shape geometry using predefined mechanical properties and external forces and ensure the object’s shape aligns with the target image when in static equilibrium.

**Weaknesses:**

In this paper, the authors primarily apply the physical compatibility optimization framework to enhance the physical attributes of 3D models obtained from existing methods rather than reconstructing them from a single image. Therefore, I think the title "Physically Compatible 3D Object Modeling from a Single Image" may not be suitable, as the focus lies on enhancing physically compatible modeling of 3D objects derived from off-the-shelf methods.

**Questions:**

1. All $\mathbf{x}\_{static}$ may be $\mathbf{X}_{static}$ as the defination in line 89.
2. In Some cases discussed in the paper, like the flamingo standing on one leg, should it be standable even after optimization? I think it should not be standable under gravity.
3. Why do you evaluate the different off-the-shelf methods using connected components when this cannot demonstrate the superiority of your proposed method?

**Limitations:**

The authors have discussed the limitations and potential negative societal impact.

---

> ### Author Rebuttal · Authors · 2024-08-06
>
> We appreciate the reviewer's effort in reading and evaluating our work carefully!
>
> **Rebuttal Outline:**
>
> 1. **[Discussion] (W1)** Title.
> 2. **[Adjustment] (Q1)** $\textbf{x}_{static}$.
> 3. **[Clarification] (Q2)** 3D Printing Results of Flamingo.
> 4. **[Clarification] (Q3)** Evaluation on the number of Connected Components.
>
> Please find below our detailed responses to your comments and concerns. We hope these responses assist you in finalizing your assessment of our manuscript's rating.
>
> ---
>
> **1. (W1) The title may not be suitable, as the focus lies on enhancing physically compatible modeling of 3D objects derived from off-the-shelf methods.**
>
> We appreciate your observation regarding the scope of our work and the paper’s title.
>
> The title "Physically Compatible 3D Object Modeling from a Single Image" was chosen to highlight our method's unique process: starting with a single image and constructing 3D object under physical compatibility constraints. Although our approach utilizes existing 3D reconstruction techniques, the core of our contribution lies in the introduction and application of a novel optimization framework that ensures physical compatibility. We believe that retaining the current title accurately reflects the novelty of our contribution.
>
> However, we are open to considering alternative titles such as "Enhancing Physical Compatibility of 3D Object Modeling from Single Images" if it would provide clearer communication of our work's scope and impact.
>
> ---
>
> **2. (Q1) All $x_{static}$ may be $X_{static}$ as the definition in line 89.**
>
> Thank you for your attentive reading! In our manuscript, we adhere to the conventional notation system used in physical simulation and finite element analysis [1], where the rest shape is denoted by capital letters $\textbf{X}$ and the static shape by lowercase letters $\textbf{x}$. To resolve the confusion, we will adjust the notation in line 89 to use $\textbf{x}$ consistently. Additionally, we will include a clarifying statement in the text to explicitly define that $\textbf{X}$ represents the rest shape.
>
> ---
>
> **3. (Q2) For the flamingo standing on one leg, should it be standable even after optimization?**
>
> Our optimization indeed successfully enables the flamingo model to stand on one leg. To address your concern, we've included a supplementary PDF in this rebuttal, showing the 3D-printed results of the standing flamingo after optimization from various camera angles. We hope this additional evidence clarifies the effectiveness of our method.
>
> ---
>
> **4. (Q3) Why do you evaluate the different off-the-shelf methods using connected components?**
>
> Thank you for the question. While the metric of connected components may not directly demonstrate the superiority of our proposed method, it remains a critical dimension for evaluating physical compatibility. This metric effectively captures the integrity of a 3D object, which is an essential attribute for assessing the physical realism of 3D models, since disconnected parts fall apart and thus naturally invalidate the definition of "physical compatibility". Including it provides valuable insights into the baseline performance of existing off-the-shelf methods with respect to physical compatibility.
>
> ---
> **References**
>
> [1] Eftychios Sifakis, and Jernej Barbic. "FEM simulation of 3D deformable solids: A practitioner's guide to theory, discretization, and model reduction." ACM SIGGRAPH 2012 courses. 2012. 1-50.
>
> ---
> We appreciate your time! Thank you so much!
>
> Best regards,
>
> Authors

---

> > ### Comment · Reviewer_H9KU · 2024-08-11
> >
> > Thanks for your efforts.
> >
> > For W1, I think your response did not fully address my concerns. I want to point out that your work is not directly related to 3D object modeling from a single image. The proposed revised title "Enhancing Physical Compatibility of 3D Object Modeling from Single Images" and your original title have no obvious difference.

---

> ### Author Response · Authors · 2024-08-11
> **Reply**
>
> Thank you for your feedback. Based on your suggestions, we will change the title to ‘Physically Compatible 3D Object Modeling.’ We are also open to any other title suggestions from the reviewer and will adjust the paper title based on the meta-review if necessary.
>
> Please feel free to let us know if there's anything else we can provide to encourage you to increase the score.
>
> Best,
>
> Authors

---

### Official Review · Reviewer_cqis · 2024-07-13

**Soundness:** 3
**Presentation:** 3
**Contribution:** 3
**Rating:** 8
**Confidence:** 3

**Summary:**

The paper presents a 3D mesh optimization framework to ensure physical plausible 3D object reconstructions from a single image. These 3D reconstructions should conform with global (e.g. gravity) and user-defined constraints (material stiffness) as well as match with a target image of the reconstructed, simulated object. The reconstructed mesh is considered as a solid object.

The method is combined with 5 existing single-image 3D object reconstruction methods, which are evaluated on a collection of 100 Objaverse samples. Several additional metrics are introduced to test for the physical 'compatibility' of the reconstructions: mean stress, connected components, standability and the difference to the reference view.

**Strengths:**

The motivation for physically plausible reconstruction is clearly stated. Ensuring physical constraints of reconstructed 3D meshes allows for further downstream applications, such as 3D printing, hence it is an important aspect to consider.

The writing and quality of the figures (including the video) is of very high quality. The set of metrics is well chosen and the paper includes very detailed descriptions of the method. Its effectiveness is demonstrated using real-world 3D printed results with and without the proposed optimization.

**Weaknesses:**

- While I can understand the space constraints of the submission format, the main paper should still include a complete and comprehensive overview of related work instead of deferring it to the appendix.

**Questions:**

- Figure 3: The plot for TetSphere show minimal improvements using the proposed optimization. Could the authors elaborate on this?

**Limitations:**

The checklist is complete and the paper discusses its limitations.

---

> ### Author Rebuttal · Authors · 2024-08-06
>
> We thank the reviewer for the constructive questions.
>
> **Rebuttal Outline:**
>
> 1. **[Adjustment] (W1)** Regarding the placement of the Related Work section.
> 2. **[Question] (Q1)** Elaboration on results of TetSphere.
>
> Please find below our detailed responses to your comments and concerns.
>
> ---
>
> **1. (W1) The main paper should still include a complete and comprehensive overview of related work instead of deferring it to the appendix.**
>
> We will adjust the manuscript to include the Related Work section within the main text. We plan to move Figure 5 to the Appendix and compress Figure 4 to free up sufficient space to accommodate the Related Work section within the page limits.
>
> ---
>
> **2. (Q1) Figure 3: The plot for TetSphere shows minimal improvements using the proposed optimization. Could the authors elaborate on this?**
>
> The modest improvement observed for TetSphere is attributed to its use of an explicit volumetric representation. As discussed in TetSphere's original paper, this representation inherently enhances robustness to slender shapes and thin structures. Unlike the other four methods we evaluated, which do not utilize explicit volumetric representations and thus are more vulnerable to fractures in thin structures, TetSphere maintains structural integrity. Nevertheless, our optimization method still enhances the robustness of TetSphere to fractures, as demonstrated in Figure 3.
>
> ---
>
> Please let us know if you have any further questions. We appreciate your time and effort. Thank you!
>
> Best regards,
>
> Authors

---

### Author Rebuttal · Authors · 2024-08-06

# General Response

We sincerely appreciate the detailed reviews and the thoughtful feedback provided by all reviewers and the Area Chair. In addition to addressing specific comments from each reviewer, we would like to outline our primary contributions.

- **[Motivation]** We tackle a critical issue in 3D generative modeling -- enhancing the **physical compatibility** of generated objects, a vital aspect often overlooked in contemporary research [cqis, 91LB].
- **[Methodology]** Our approach is **novel** and significantly improves the **versatility and applicability** of 3D reconstruction methods [H9KU, 91LB, XLVo], evident through the integration of physical constraints within the generative process.
- **[Experiments]** We conducted comprehensive experiments that not only demonstrate the **robustness and versatility** of our method but also extend its applicability to **real-world scenarios** like 3D printing [cqis, 91LB].
- **[Presentation]** The manuscript is recognized for **its clarity and well-written quality** [cqis, H9KU].

We hope our responses address all reviewers' concerns and help improve the review scores. We thank all reviewers and the AC again for their time and efforts!

---

### Decision · Program_Chairs · 2024-09-25

**Decision:**

Accept (spotlight)

**Comment:**

All the reviewers are positive and recommending acceptance of the submission. The AC agrees with the decision. Please read the weakness comments carefully and prepare the camera ready.